# Short-Term Effects of Escalating Doses of Cholecalciferol on FGF23 and 24,25(OH)_2_ Vitamin D Levels: A Preliminary Investigation

**DOI:** 10.3390/nu16213600

**Published:** 2024-10-23

**Authors:** Jessica Pepe, Luciano Colangelo, Roberta Pilotto, Viviana De Martino, Carla Ferrara, Alfredo Scillitani, Mirella Cilli, Salvatore Minisola, Ravinder Singh, Cristiana Cipriani

**Affiliations:** 1Department of Clinical, Internal, Anesthesiology and Cardiovascular Sciences (SCIAC), “Sapienza” University of Rome, 00161 Rome, Italy; luciano.colangelo@uniroma1.it (L.C.); robertapilotto@virgilio.it (R.P.); viviana.demartino@uniroma1.it (V.D.M.); mirella.cilli@uniroma1.it (M.C.); salvatore.minisola@uniroma1.it (S.M.); cristiana.cipriani@gmail.com (C.C.); 2Istituto Superiore di Sanità, 00161 Rome, Italy; carla.ferrara@iss.it; 3Unit of Endocrinology, Ospedale Casa Sollievo della Sofferenza, IRCCS, San Giovanni Rotondo, 71013 Foggia, Italy; alfredo.scillitani@gmail.com; 4Department of Laboratory Medicine and Pathology, Mayo Clinic, Rochester, MN 55905, USA; singh.ravinder@mayo.edu

**Keywords:** vitamin D, 24,25(OH)_2_D, FGF23, cholecalciferol

## Abstract

Background: There are few and controversial results on 24,25(OH)_2_D and FGF23 acute changes following supplementation with cholecalciferol. Methods: Twenty-seven subjects with 25(OH)D < 30 ng/mL were randomized into three groups to receive a single oral dose of 25,000 I.U. or 600,000 I.U. of cholecalciferol or placebo, respectively. We measured 25(OH)D, 1,25(OH)_2_D, 24,25(OH)_2_D, and FGF23 levels at baseline and after 72 h. The 1,25(OH)_2_D/25(OH)D, 1,25(OH)_2_D/24,25(OH)_2_D, and 24,25(OH)_2_D/25(OH)D ratios were also calculated. Results: There was an increase in 25(OH)D and 1,25 (OH)_2_D following both doses of cholecalciferol. In the group administered 600,000 I.U., there was a significant increase in the delta changes in 25(OH)D and 1,25(OH)_2_D compared to the placebo and in the delta 24,25(OH)D_2_ compared to the placebo and 25,000 I.U. groups (all *p* < 0.05). A decrease in both the 1,25(OH)_2_D/25(OH)D and 1,25(OH)_2_D/24,25(OH)_2_D ratio (all *p* < 0.05) was observed in the 600,000 I.U. group. FGF23 values significantly increased only in the group administered 600,000 I.U. Conclusions: 25(OH)D and 1,25(OH)D levels significantly increased following 600,000 IU cholecalciferol administration compared to 25,000 I.U. and placebo. Following the massive administration of cholecalciferol, the CYP24A1 enzyme is actively involved in catabolism, thus, avoiding toxic effects.

## 1. Introduction

There are many ways to replenish the human body with vitamin D. The most physiological one is represented by ultraviolet B (UVB) radiation. UVB is in the region of the ultraviolet spectrum, which extends from about 280 to 320 nm in wavelength. Everything below 290 is absorbed by the stratosphere; therefore, for the natural light range the string from 290 is relevant. Erythemally weighted UVB above 315 nm is irrelevant in D3 formation, thus, the biological effect of this spectrum is through a different mechanism. The peak of production is around 295 nm. It has been shown that many biological effects of UVB cannot be replaced by oral delivery [1]. However, some concerns have been raised, since UVB radiation is primarily responsible for sunburn, aging of the skin, and the development of skin cancer. In this context, it is important to highlight the discovery of novel pathways of vitamin D3 initiation, for example, by CYP11A1 and/or CYP27A1. The generated metabolites among other things, enhance the protection of skin against DNA damage and oxidative stress, thus, protecting its integrity [2]. Furthermore, it has also been hypothesized that these hydroxy-metabolites might play a role in some so-called non-skeletal effects of vitamin D, including systemic diseases such as cancer, autoimmune and inflammatory disorders, atherosclerosis, and many metabolic diseases. Under most living conditions, approximately 75 to 90% of this pro-hormone is produced in the skin after UVB exposure. However, many factors can influence production, the most important being latitude, season, and time of day as well as ozone pollution in the atmosphere. Concerning pollution, it has been shown that the more an area is polluted the more the passage of UVB is diminished, consequently impairing the synthesis of vitamin D. Another physiological possibility is represented by foods and other nutritional sources [3]. However, only a small amount of vitamin D is available from nutritional sources since very few foods naturally contain vitamin D. The flesh of fatty fish (including, trout, salmon, tuna, and mackerel) and fish oils are among the best sources. Beef liver, egg yolks, and cheese have small amounts of vitamin D, primarily in the form of vitamin D3 and its metabolite 25(OH)D3. Mushrooms provide variable amounts of vitamin D2 [4]. The third option is represented by enriching foods with vitamin D or 25(OH)D. Several countries (for example, USA, India, Finland, Canada, and other nations in the Middle East and North African regions) mandate the addition of vitamin D to various foods. However, many other countries are reluctant to accept this approach, even though this could substitute pharmacological vitamin D supplementation without any harmful effects. Finally, the last option is represented by pharmacological supplementation [5]. Concerning the last option, there are a number of commercial products that can be utilized that basically recapitulate the biosynthetic pathway of vitamin D from its origin to the final metabolite. The most utilized of these products are ergocalciferol, cholecalciferol, calcifediol, 1α(OH)D, and calcitriol, even though their utilization is markedly different in European, Asian, and American countries. Other compounds, for example, paricalcitol, are utilized in specific clinical conditions, such as in chronic kidney disease [4].

Cholecalciferol (vitamin D3) is the most commercialized product in Europe. Its supplementation has been proposed as one of the modalities to overcome the widespread prevalence of vitamin D insufficiency in the general population [6,7]. To date, most studies have mainly focused on the ability of different schedules of cholecalciferol administration (daily, weekly, fortnightly, monthly, and even every six months) to reach the vitamin D sufficiency evaluated by measuring circulating 25(OH)D levels, mainly on long-term basis. However, there is little data on the effect of different doses of cholecalciferol on their final active [i.e., 1,25(OH)_2_D] and catabolic [i.e., 24,25(OH)_2_D] derivatives [8]. Furthermore, the majority of the studies evaluate the production of 24,25(OH)D over the long term, i.e., weeks or months after vitamin D supplementation of vitamin D deficient population [9,10,11,12,13]. However, to the best of our knowledge, there are no data on the early changes in 24,25(OH)_2_D nor of 24,25(OH)_2_D/25(OH)D ratio (a marker of vitamin D metabolism) after different doses of cholecalciferol supplementation. In this context, it should be noted that some data seem to suggest that 24,25(OH)_2_D cannot be considered only as an inert metabolite. Indeed, it has been implicated in many biological processes independent of the action of 1,25(OH)_2_D.

*CYP24A1*, the enzyme needed for the catabolism of both 25(OH)D and 1,25(OH)_2_D into presumably inactive carboxylic acids, is also influenced by fibroblast growth factor (FGF23) [13]. In particular, FGF23 suppresses the expression of the 1α-hydroxylase enzyme in the kidney and induces the expression of 25-hydroxyvitamin D 24-hydroxylase enzyme, resulting in the decreased conversion of 25(OH)D into 1,25(OH)_2_D and increased formation of inactive carboxylic acids [14]. A recent meta-analysis showed that vitamin D supplementation leads to a significant increase in serum intact FGF23 among vitamin D-deficient patients [15]. Authors considered studies employing both daily doses and boluses of vitamin D up to 300,000 international units (I.U.), with a follow-up from eight weeks to three months. Finally, according to some authors, a more extensive metabolome profile of vitamin D, including different vitamin D metabolic ratios, could offer insights to both evaluate genetic and acquired diseases of calcium and phosphorus metabolism and better elucidate metabolic pathways [16,17].

Vitamin D metabolism strictly regulates not only calcium but also phosphate metabolism. For example, if the dietary intake of phosphate is low, absorption is mostly active and mediated through 1,25-dihydroxyvitamin D. This final metabolite is able to induce the expression of NPT2b (sodium-phosphate transport protein 2B), which is a sodium-phosphate cotransporter on the apical membrane of intestinal epithelial cells. In the kidneys, 1,25(OH)_2_D can stimulate NPT2a to increase phosphate reabsorption. NPT2c is typically increased during a low-phosphate diet but reduced in response to a high-phosphate diet. 1,25(OH)_2_D stimulates both the functional role and biosynthesis of NPT2c [18].

Therefore, the aim of this study was to evaluate the short-term effect (i.e., 72 h) of a small and large oral dose of cholecalciferol on FGF23 and 24,25(OH)_2_D in vitamin D deficient patients, together with the evaluation of three vitamin D metabolic ratios (1,25(OH)_2_D/25(OH)D, 1,25(OH)_2_D/24,25(OH)2D, and 24,25(OH)_2_D/25(OH)D).

## 2. Materials and Methods

### 2.1. Study Design, Recruitment, Inclusion and Exclusion Criteria

Between January and March 2015, we initially investigated the first 70 Caucasian postmenopausal women and men (age range 40–90 years), routinely followed as outpatients at our Department of Internal Medicine. Potentially eligible individuals were subjected to an assessment visit, consisting of medical history, clinical examination (vitals, weight, height, and physical examination, including blood pressure and pulse rate), and biochemical evaluations. The aim was the enrollment of 27 patients, satisfying the inclusion [25(OH)D levels < 30 ng/mL] and exclusion criteria, as a preliminary investigation to define future protocols on the basis of the results obtained. Therefore, as an exploratory analysis, a sample size calculation was not estimated. The exclusion criteria were as follows: estimated creatinine clearance by Cockcroft–Gault formula < 60 mL/min, not suffering from diseases such as neoplasia, hepatitis C and/or B virus co-infection and other liver diseases, skeletal diseases (such as primary hyperparathyroidism, Paget’s disease or osteoporosis under treatment), severe water and electrolyte disturbances, malabsorption, acute or chronic renal failure, or any serious systemic diseases that could affect the interpretation of the data. Among 70 patients screened, we report the data from 27 enrolled patients whose vitamin D levels were below 30 ng/mL. None of them were taking drugs affecting bone metabolism or calcium or vitamin D supplements. Patients were randomized into three groups of 9 subjects each to receive a single oral dose of 25,000 or 600,000 I.U. of cholecalciferol or the placebo in an open-label modality. All participants were randomly assigned by a computer-based randomization system. Fasting blood samples were collected at the baseline and 72 h after different doses of cholecalciferol or placebo administration, in all participants. One subject in the 25,000 IU was lost in the follow-up. There were no changes in dietary habits that could have influenced the results during the period of observation.

### 2.2. Ethical Considerations

The study was approved by our local Ethic Committees and written informed consents have been preliminarily obtained by each subject.

### 2.3. Biochemistry

Following the immediate analysis of serum ionized calcium, the blood samples were centrifuged at a speed of 3000 rpm for 10 min and the serum was frozen at −80 °C. Assays of the analytes were performed in a single batch at the end of the study. We measured the serum intact parathyroid hormone (PTH), 25-hydroxyvitamin D [25(OH)D], 1,25-dihydroxyvitamin D [1,25(OH)_2_D], and intact FGF23 (FGF23), as previously described [19,20]. Briefly, the serum ionized calcium (Ca^2+^) was determined using an ion-selective electrode with the fully automated biochemical analyzer NOVA 8 (Nova Biomedical, Waltham, MA, USA). Serum 25-hydroxyvitamin D [25(OH)D] levels and 1,25-dihydroxy-vitamin D [1,25(OH)_2_D] were analyzed by radioimmunoassay (RIA, Immunodiagnostic Systems Inc. (IDS Inc.), Fountain Hills, AZ, USA). The sensitivity of the 25(OH)D assay is 1.5 ng/mL, and the intra-assay and inter-assay CVs are below 10.8% and 9.4%, respectively, and the sensitivity of the 1,25(OH)_2_D assay is 4.6 pg/mL, and the intra-assay and inter-assay CVs are below 6.3% and 9.5%, respectively. The serum levels of PTH were analyzed by immunoradiometric assay (IRMA N-TACT PTH SP, Diasorin, Saluggia, Italy) according to the manufacturer’s instructions. The sensitivity of this assay is 0.07 pg/mL, and the intra-assay and inter-assay CVs are below 2.6% and 4.3%, respectively. FGF23 was determined by a human intact FGF23 ELISA kit (Kainos Laboratories, Tokyo, Japan), and the intra- and inter-assay CVs were less than 8% and sensitivity was 3 pg/mL [21]. 24,25(OH)_2_ vitamin D levels were measured by liquid chromatography–tandem mass spectrometry (LC–MS/MS) at the Mayo Clinic. More specifically, 10 g/mL solution of 24R,25(OH)_2_D3 and 24R,25(OH)_2_D2 (Medical Isotopes) in 100% ethanol was prepared in the laboratory. Concentrations of 0, 0.1, 0.5, 1, 5, 10, and 25 ng/mL [0, 0.2, 1.2, 2.4, 12.0, 24.0, and 60.0 nmol/L for 24,25(OH)_2_D3 and 0, 0.2, 1.2, 2.3, 11.7, 23.3, and 58.3 nmol/L for 24,25(OH)_2_D2] for calibrators; QC samples (prepared in stripped serum) were at a concentration of 0.6, 5.5, and 14 ng/mL [1.4, 13.2, and 33.6 nmol/L for 24,25(OH)_2_D3; 1.4, 12.8, and 32.7 nmol/L for 24,25(OH)_2_D2]. The internal standard of 50 µL [1.25 ng (0.003 nmol)] deuterated 24,25(OH)_2_D3 and deuterated 24,25(OH)_2_D2 (Toronto Research Chemicals, North York, ON, Canada) to 500 µL serum followed by 500 µL hydrochloric acid (0.2 N) 15 min later. Extraction was performed with BondElut (C18, 250 mg, 6 mL) cartridge (Varian Instruments, Palo Alto, CA, USA), which was washed once with 2 mL of 70:30 methanol:water (1.4 mL methanol:0.6 mL water) and then with 2 mL of 90:10 hexane:methylene chloride (1.8 mLhexane:0.2 mL methylene chloride). Samples were eluted with 2 mL of 90:10 hexane:isopropyl alcohol (1.8 mL hexane:0.2 mL isopropyl alcohol). Eluents were dried and derivatized with 4-phenyl-1,2,4,triazoline-3,5-dione (PTAD) [Sigma, Kawasaki, Japan; 250 µL of 200 µg/mL (285 µmol/L) solution in acetonitrile]. Analytes were separated by column chromatography at a flow rate of 0.25 mL/min on an Agilent XDB-C8 (Agilent, Santa Clara, CA, USA), 2.1 50 mm column over 7.25 min with a methanol-H_2_O-ammonium formate (1 mM) linear gradient (60–95%). Mass spectrometry was performed in the multiple reaction monitoring mode on a AB Sciex 5500 mass spectrometer with Analyst 1.6.2 software (AB Sciex, Framingham, MA, USA) for data acquisition and analysis. We also calculated the following derived vitamin D metabolic ratios: (a) 1,25(OH)_2_D/25(OH)D ratio, mainly representing the 1α-hydroxylase activity, an enzyme encoded by *CYP27B1* gene in the kidney; (b) 1,25(OH)_2_D/24,25(OH)_2_D ratio as an indicator of vitamin D status; (c) 24,25(OH)_2_D/25(OH)D as a ratio mediated by the *CYP24A1* gene that encodes the enzyme 24-hydroxylase, which catalyzes the conversion of 25(OH)D into 24,25(OH)_2_D, thus, representing the catabolic activity.

### 2.4. Safety Endpoints

Safety and tolerability assessments consisted of adverse events (AEs), clinical laboratory (clinical chemistry, haematology, coagulation, and urinalysis), vital signs (systolic and diastolic blood pressure, heart rate, and body temperature), 12-lead electrocardiogram, body weight, falls, and physical examination.

### 2.5. Statistical Analyses

Patient’s biochemical measures according to time from vitamin D or placebo administration are reported as means ± SD. The Kolmogorov–Smirnov test was used to evaluate the normality of the distributions. All parameters were normally distributed with the exception of ionized calcium, BMI, 1,25(OH)_2_D/25(OH)D, 1,25(OH)_2_D/24,25(OH)_2_D, 24,25(OH)_2_D/25(OH)D, and FGF23. Since the above-mentioned distribution parameters did not follow the Gaussian curve, non-parametric tests were carried out as follows: Kruskal–Wallis ANOVA for comparison between groups (Table 1) and pairwise comparisons using the Wilcoxon test was made in each group (Table 2). For the other variables, ANOVA was performed to evaluate the delta changes in indices in the groups throughout the study (Figure 1). Adjustments for multiple comparisons (Bonferroni) were also made when the ANOVA test was statistically significant. Fisher’s exact two-tailed test were used for categorical variables comparisons between groups. Bonferroni correction was applied to account for multiple tests. The significance was set at a *p* value of <0.05. The statistical analyses were carried out using the SSPS version 21.

## 3. Results

We enrolled 12 men and 14 women in the study. The mean (± SD) age of the sample studied was 73.8 ± 12.7 years, while the mean body mass index was 25.3 ± 4.1 kg/m^2^. Table 1 reports the main anthropometric and biochemical parameters of the patients investigated. There were no significant differences in the distribution of females or males among the three groups investigated, as well as in the anthropometric and basal biochemical parameters.

There was a significant increase in the mean 25(OH)D levels with respect to the basal values both in the 25,000 and 600,000 I.U. groups but not in the placebo group on the third day following cholecalciferol or placebo supplementation, respectively (Table 2). If we consider 30 ng/mL value as a sufficiency threshold for vitamin D, none of the subjects in the group supplemented with 25,000 I.U. reached this target after 72 h, while seven out of nine patients in the 600,000 I.U. group reached this target. The number of patients reaching sufficiency in the last group (i.e., 600,000 I.U.) was significantly higher compared to those supplemented with 25,000 I.U. of cholecalciferol (*p* = 0.004).

In the 600,000 I.U. group, the mean 25(OH)D values increased from 16.55 ± 13.1 to 58.6 ± 29.4 ng/mL (*p* < 0.05), while in the 25.000 I.U. group mean 25(OH)D values increased from 12.2 ± 8.8 to 19.9 ± 11.3 ng/mL (all *p* < 0.05). A significant difference was observed among the three groups (ANOVA *p* = 0.02) (Figure 1 left panel) when comparing the delta changes in 25(OH)D from the baseline after vitamin D supplementation. The difference in the delta changes in 25(OH)D between the controls and 600,000 I.U. group persists also after Bonferroni correction.

Delta changes (72 h value–basal value) of 25(OH)D vitamin D (left panel), 1,25(OH)_2_D dihydroxy vitamin D (middle panel), and 24,25(OH)_2_D dihydroxy vitamin D (right panel) in the controls and in the patients supplemented with 25,000 and 600,00 I.U. of cholecalciferol after 72 h are reported in Figure 1.

In the case of 1,25(OH)_2_D serum levels, we found a significant absolute increase (all *p* < 0.05) after 72 h both in the groups supplemented with 25,000 I.U. and 600,000 I.U. of cholecalciferol. Differences in the mean delta changes among groups were significant (ANOVA *p* = 0.009), as shown in Figure 1 middle panel. After Bonferroni correction, a statistically significance difference in the mean delta increase was observed only in the group supplemented with 600.000 I.U. compared to the control group.

The administration of 600,000 I.U. of cholecalciferol led to an increase in the mean delta 24,25(OH)_2_D changes that were significantly higher compared to the other groups, (ANOVA *p* = 0.008), as shown in Figure 1 right panel. In particular, the differences between the delta changes in those supplemented with 600,000 I.U. were 2.11 ± 0.73 compared to the controls (Bonferroni corrected *p* = 0.03) and 2.56 ± 0.76 compared to those supplemented with 25,000 (Bonferroni corrected *p* = 0.01).

We found a significant mean decrease in the ratio of 1,25(OH)_2_D/25(OH)D and 1,25(OH)_2_D/24,25(OH)_2_D (both *p* < 0.05), but not of 24,25(OH)_2_D/25(OH)D (Table 2) ratio at 72 h, only in the group supplemented with 600,000 I.U. of cholecalciferol.

Considering the subjects investigated as a whole, we found a significant correlation between the delta 25(OH)D and delta 24,25(OH)_2_D (r = 0.47, *p* = 0.02), and a significant positive correlation between the absolute value of 25(OH)D at baseline and 24,25(OH)_2_D after 72 h (r = 0.56, *p* = 0.04). Moreover, a significant positive correlation between the absolute value of 1,25(OH)_2_D at baseline and 24,25(OH)_2_D after 72 h was observed (r = 0.56, *p* = 0.005).

Concerning PTH, we did not find significant mean changes after 72 h following cholecalciferol supplementation, even though there was a decreasing trend in those supplemented with 600,000 I.U. Interestingly, in the last group (i.e., in those administered 600,000 I.U) a significant increase in the mean serum ionized calcium (1.24 ± 0.03 vs. 1.28 ± 0.16 mmol/L, *p* < 0.05) and mean serum FGF23 were observed (48.2 ± 18.7 vs. 68.9 ± 34.2, *p* < 0.05). However, delta changes among the three groups for these two variables did not reach statistical significance.

No case of severe hypercalcemia was observed, nor any clinical symptom was reported by the subjects during the 3 days after the administration of cholecalciferol or placebo.

## 4. Discussion

To the best of our knowledge, this is the first investigation showing early changes in 24,25(OH)_2_D and FGF23 values after two different dosing regimens of cholecalciferol in ambulatory patients with vitamin D insufficiency attending an internal medicine department. As expected [22,23,24], we found that the mean increase in 25(OH)D and 1,25(OH)_2_D was more pronounced following 600,000 I.U. administration. This last dose also determined a very mild increase in the mean values of the serum ionized calcium.

Concerning changes in 24,25(OH)_2_D values, we found that there were significant mean increases following the bolus dose (i.e., 600,000 I.U.) with respect to both the control subjects and those administered 25,000 I.U. of cholecalciferol. Previous studies have shown changes in 24,25(OH)_2_D on a long-term basis, i.e., 30 days after a single bolus (by the oral or intramuscular route) of 600,000 I.U. [25], 4 weeks following an oral bolus dose of 100,000 I.U. [12], 8 weeks following 28,000 I.U. once per week [11], or 12 weeks following a daily dose of 20 mcg [26]. However, in a study carried out on healthy young subjects administered different doses of cholecalciferol (10,000 I.U./day for 8 weeks followed by 1000 I.U./day for 4 weeks; or 50,000 I.U./week for 12 weeks, or 100,000 I.U. every other week for 12 weeks), 24,25(OH)_2_D values did not change over time when measured at 28, 53, 84 and 112 days [27]. These inconsistencies may be ascribed to many factors, including assay procedure, dose and time interval of cholecalciferol administration, age, and comorbidities.

Our investigation fills a gap in knowledge in this field. Indeed, as outlined in a recent editorial [28], it is important to measure the levels of vitamin D metabolites in patients administered bolus doses of vitamin D3. This is particularly important in view of some previous studies warning against the use of bolus doses of cholecalciferol in vitamin D-repleted subjects [29]. These huge doses would favor falls with consequent increased risk of fractures, so that they should be avoided [30]. Even though the mechanisms linking bolus cholecalciferol administration to falls have not been completely elucidated, a potential negative effect of increased 24,25(OH)_2_D has been called into question. We did not register any falls during the three-day observation period; however, the number of subjects investigated was very small so that we cannot draw any definitive conclusion. In addition, our study was carried out in vitamin D insufficient subjects that is a completely different clinical condition. In this context, the role of 24,25(OH)_2_D with respect to skeletal health has not been completely clarified. For example, some studies show that the administration of 10,000 I.U. of cholecalciferol for 36 months has negative effects on radial total bone mineral density [31]. However, previous studies have shown that the administration of huge doses of 24,25(OH)D to mice with X-linked hypophosphatemia [32] or patients with kidney failure [33] did not result in an increased osteoclast surface area when evaluated from a histological point of view.

The changes we found following the administration of the largest dose of cholecalciferol (600,000 I.U), are in line with our current understanding of vitamin D metabolism. Indeed, the administration of a huge dose of cholecalciferol results in a more consistent increase in 25(OH)D than 1,25(OH)_2_D (and, therefore, a decrease in the 1,25/25 ratio). The massive increase in both metabolites triggers the expression of mitochondrial CYP24A1, the principal enzyme that catalyzes the metabolic clearance of 25(OH)D into 24,25-dihydroxyvitamin D, (thus, also determining the reduction of 1,25/24,25 ratio). As a result, the ratio between 24,25/25 does not change.

In our study, we found no differences in 24,25(OH)_2_ D/25(OH)D ratio, in line with what was observed in previous studies [8,10] but contrary to what was observed in Tang’s study [34].

Changes in the serum calcium and PTH values following the bolus dose are in line with our previous paper showing similar biochemical findings [25]. However, also in this case, we must acknowledge inconsistent results in the literature regarding parathyroid hormone behavior following cholecalciferol supplementation [22,34,35]. The same confounding factors previously addressed might underlie different results.

Concerning serum FGF23 levels, we showed significant mean increases after 72 h of 600,000 I.U. of cholecalciferol, while no changes were detected following 25,000 I.U. Indeed, our results are in line with data from a recent meta-analysis documenting an associated increase in serum FGF23 with cholecalciferol supplementation after a longer period of time following vitamin D supplementation with different dosages [15]. A number of possible mechanisms can underlie this increase. Among these, is the possibility that repletion of vitamin D might lead to an increase in intracellular 1,25(OH)_2_D concentration in the osteocyte, which would then increase the synthesis and secretion of FGF23. Also, the systemic increase in 1,25(OH)_2_D may be counteracted by the increase in FGF23 by downregulating the expression of the vitamin D-activating enzyme cytochrome P450 subfamily 27 polypeptide 1 (CYP27B1; also known as mitochondrial 25-hydroxyvitamin D1 α-hydroxylase) and by upregulating the vitamin D-degrading enzyme CYP24A1.

### Strengths and Limitations

Our study has some limitations worth mentioning. First, we only focused on acute changes, since this information is still missing in the literature, as previously reported [28]. A longer treatment period may have revealed additional information on vitamin D metabolism. A follow-up investigation is ongoing in order to meet these needs. Second, we might also have considered the effects of other doses of vitamin D. However, at the time we carried out this investigation, the doses we employed were the most frequently used in clinical practice in our country. However, following a paper by Sanders and coworkers [29] that utilized 500,000 I.U. annually, the utilization of huge bolus doses of vitamin D has been abandoned for the possible increased risk of falls and fractures. In this context, it should be emphasized that, even though there has been a large amount of scientific investigation, the causal link between the administration of huge doses of vitamin D and the increased risk of falls has never been elucidated. Third, we did not measure other vitamin D metabolites such as 3-epi-25(OH)D3; however, the role of this metabolite seems to be marginal in adults. Other derivatives (such as CYP111A1 derivate vitamin D) might give more information if assayed in future studies. In future research, we are also focusing on serum phosphate and possible sex differences following the administration of different doses of cholecalciferol. Finally, considering the ongoing debate on adverse events linked to bolus doses of vitamin D, we might have measured other parameters such as, for example, the biochemical markers of skeletal turnover. However, this was not the end point of our investigation.

## 5. Conclusions

In conclusion, both a single dose of 25,000 and 600,000 I.U. of cholecalciferol increase 25(OH) levels even though to a different degree. Moreover, the 600,000 I.U. dose results in a higher magnitude of conversion to the 24,25(OH)_2_D compared to the lower dosage of vitamin D supplementation. Furthermore, the highest dose results in immediate homeostatic adjustments, although they are slower with the more physiological dose. These adjustments are mainly represented by increased FGF23 secretion and CYP24A1 activity. Our study was preliminary in nature since we investigated three small groups of subjects. However, if our results can be confirmed in larger populations with different doses and compounds, they will give important clues about homeostatic adjustments following vitamin D supplementation and treatment. This is especially important in light of the discovery of new metabolic functions of previously disregarded vitamin D compounds.

## Figures and Tables

**Figure 1 nutrients-16-03600-f001:**
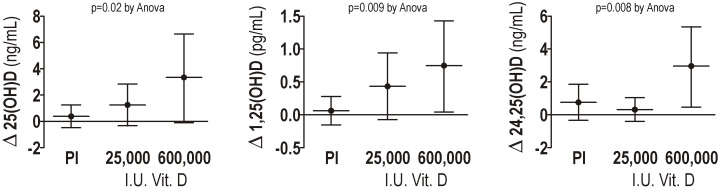
Delta changes in 25(OH)D, 1,25(OH)_2_D, and 24,25(OH)_2_D, after 72 h, in the groups studied. Pl: placebo.

**Table 1 nutrients-16-03600-t001:** Anthropometric and biochemical parameters of three groups of subjects investigated.

	Controls (n = 9)	25,000 I.U. (n = 8)	600,000 I.U. (n = 9)	*p* Values
**Age (years)**	70.4 ± 17.1	75.9 ± 10.6	75.2 ± 9.7	0.63
**BMI (Kg/m^2^)**	25.4 ± 5.2	26.2 ± 2.0	24.4 ± 4.6	0.90
**Ca^++^ (mmol/L)**	1.21 ± 0.08	1.24 ± 0.04	1.24 ± 0.03	0.50
**25(OH)D (ng/mL)**	11.5 ± 8.6	12.2 ± 8.8	16.55 ± 13.1	0.50
**1,25(OH)_2_D (pg/mL)**	34.8 ± 12.6	27.8 ± 11.3	39.8 ± 13.55	0.13
**24,25(OH)_2_D (ng/mL)**	0.50 ± 0.45	0.72 ± 0.76	0.72 ± 0.75	0.91
**PTH (pg/mL)**	55.2 ± 36.0	74.0 ± 57.7	45.0 ± 13.1	0.25
**FGF23 (ng/mL)**	40.0 ± 13.8	39.7 ± 26.9	48.2 ± 18.7	0.59
**1,25(OH)_2_D/25(OH)D**	0.004 ± 0.001	0.002 ± 0.001	0.003 ± 0.001	0.21
**1,25(OH)_2_D/24,25(OH)_2_D**	0.14 ± 0.12	0.16 ± 0.20	0.13 ± 0.10	0.60
**24,25(OH)_2_D/25(OH)D**	0.06 ± 0.08	0.11 ± 0.12	0.04 ± 0.03	0.31

**Table 2 nutrients-16-03600-t002:** Absolute changes in the biochemical parameters at 72 h following cholecalciferol or placebo supplementation. Time 0 vs. time 72 h in each group *****
*p* < 0.05.

	25,000 I.U.		600,000 I.U.		Controls	
	0	72	0	72	0	72
**Ca^++^ (mmol/L)**	1.24 ± 0.04	1.27 ± 0.04	**1.24 ± 0.03**	**1.28 ± 0.16 ***	1.21 ± 0.08	1.22 ± 0.72
**25(OH)D (ng/mL)**	**12.2 ± 8.8**	**19.9 ± 11.3 ***	**16.55 ± 13.1**	**58.6 ± 29.4 ***	11.5 ± 8.6	15.5 ± 11.9
**1,25(OH)_2_D (pg/mL)**	**27.8 ± 11.3**	**38.6 ± 15.8 ***	**39.8 ± 13.55**	**70.5 ± 17.1 ***	34.8 ± 12.6	30.3 ± 10.25
**24,25(OH)_2_D (ng/mL)**	0.72 ± 0.76	0.95 ± 0.98	0.72 ± 0.75	2.56 ± 1.52	0.50 ± 0.45	0.59 ± 0.32
**PTH (pg/mL)**	74.0 ± 57.7	71.7 ± 61.1	46.0 ± 13.1	38.7 ± 16.6	55.2 ± 36.0	59.7 ± 61.3
**FGF23 (ng/mL)**	39.7 ± 26.9	38.9 ± 18.08	**48.2 ± 18.7**	**68.9 ± 34.2 ***	40.0 ± 13.85	37.5 ± 27.3
**1,25(OH)_2_D/25(OH)D**	0.002 ± 0.001	0.002 ± 0.001	**0.003 ± 0.001**	**0.001 ± 0.001 ***	0.004 ± 0.001	0.004 ± 0.004
**1,25(OH)_2_D/24,25(OH)_2_D**	0.16 ± 0.20	0.07 ± 0.08	**0.13 ± 0.10**	**0.06 ± 0.06 ***	0.14 ± 0.12	0.06 ± 0.03
**24,25(OH)_2_D/25(OH)D**	0.11 ± 0.12	0.07 ± 0.04	0.04 ± 0.03	0.03 ± 0.01	0.06 ± 0.08	0.07 ± 0.06

## Data Availability

Some or all datasets generated during the current study are not publicly available due to privacy but are available from the corresponding author upon reasonable request.

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
