# Peer review of "Short-Term Effects of Escalating Doses of Cholecalciferol on FGF23 and 24,25(OH)2 Vitamin D Levels: A Preliminary Investigation"

_nutrients, 2024, doi:10.3390/nu16213600_

Round 1
Reviewer 1 Report
Comments and Suggestions for Authors
This is a timely paper and appropriate for the journal. However, the paper would benefit from following revisions.
I do not have any major critique as relates to the methodology and data collection. The predominant critique relates to the introduction, discussion and missing information in the limitation.
Information on UVB range involved in D3 formation requires corrections.
Everything below 290 is absorbed by the stratosphere, so for natural light range string from 290 is relevant. EUVB above 315 is irrelevant in D3 formation, thus the biological effect of this spectrum is through different mechanism. Peak of production is around 295 nm. There are many biological effects of UVB that cannot be replaced by oral delivery (Photo-neuro-immuno-endocrinology: How the ultraviolet radiation regulates the body, brain, and immune system. Proceedings of the National Academy of Sciences 2024 Vol. 121 Issue 14 Pages e2308374121; doi:10.1073/pnas.2308374121).
I am surprised that the authors do not mention CYP11A1 initiated pathways of vitamin D activation, which are biologically active and act o several alternative nuclear receptors in addition to the VDR (Biological Effects of CYP11A1-Derived Vitamin D and Lumisterol Metabolites in the Skin. J Invest Dermatol, 2024; doi:10.1016/j.jid.2024.04.022). These pathways are known for more than 20 years and represent already a textbook information (see recent edition of the Textbook on Vitamin D by Hewison et al.).
Worthy to mention is wide production of vitamin D and its derivatives in nature that potentially represent (Biol Rev Camb Philos Soc. 2024 Apr 30. doi: 10.1111/brv.13091) nutritional sources not mentioned in this paper.
In limitations please mention that you did not measure CYP11A1 hydroxyderivatives. Note, that CYP11A1 acts on unmodified D3, hydroxylation on C25 prevent CYP11A1 from recognizing D3 as substrate. Information on this you can find in FEBS J 2005 paper, cited in one of the above reviews.
The IRB protocols are approximately listed and consent has been obtained. Perhaps it is worthy that studies were approved by IRB in the beginning of Methodology.
There are typographical errors and errors in formatting to be corrected.
Author Response
REVIEWER 1
This is a timely paper and appropriate for the journal. However, the paper would benefit from following revisions.
Dear Reviewer,
Thanks for appreciating our manuscript that has been revised according to your and other referees’ suggestions.
I do not have any major critique as relates to the methodology and data collection. The predominant critique relates to the introduction, discussion and missing information in the limitation.
We tried to address your constructive criticism as you can see in the responses detailed below.
Information on UVB range involved in D3 formation requires corrections.
Everything below 290 is absorbed by the stratosphere, so for natural light range string from 290 is relevant. EUVB above 315 is irrelevant in D3 formation, thus the biological effect of this spectrum is through different mechanism. Peak of production is around 295 nm. There are many biological effects of UVB that cannot be replaced by oral delivery (Photo-neuro-immuno-endocrinology: How the ultraviolet radiation regulates the body, brain, and immune system. Proceedings of the National Academy of Sciences 2024 Vol. 121 Issue 14 Pages e2308374121; doi:10.1073/pnas.2308374121).
Many thanks for this important information that has now been fully incorporated into the manuscript.
I am surprised that the authors do not mention CYP11A1 initiated pathways of vitamin D activation, which are biologically active and act o several alternative nuclear receptors in addition to the VDR (Biological Effects of CYP11A1-Derived Vitamin D and Lumisterol Metabolites in the Skin. J Invest Dermatol, 2024; doi:10.1016/j.jid.2024.04.022). These pathways are known for more than 20 years and represent already a textbook information (see recent edition of the Textbook on Vitamin D by Hewison et al.).
Once again, your suggestion has been fully integrated in the revised version of the manuscript.
Worthy to mention is wide production of vitamin D and its derivatives in nature that potentially represent (Biol Rev Camb Philos Soc. 2024 Apr 30. doi: 10.1111/brv.13091) nutritional sources not mentioned in this paper.
This information has been included in the manuscript.
In limitations please mention that you did not measure CYP11A1 hydroxyderivatives. Note, that CYP11A1 acts on unmodified D3, hydroxylation on C25 prevent CYP11A1 from recognizing D3 as substrate. Information on this you can find in FEBS J 2005 paper, cited in one of the above reviews.
We added among limitations of the study, the fact that we did not measure the CYP11A1 hydroxyderivatives.
The IRB protocols are approximately listed and consent has been obtained. Perhaps it is worthy that studies were approved by IRB in the beginning of Methodology.
Thanks for your suggestion that has been followed, adding this information in the Materials and Methods section.
There are typographical errors and errors in formatting to be corrected.
Thanks once again. The manuscript has been reviewed by a native-speaking scientist.
We hope that having satisfied all the issues raised you can consider the manuscript suitable for definitive publication.
Reviewer 2 Report
Comments and Suggestions for Authors
This clinical study investigates cholecalciferol in more physiology dose and bolus dose on the serum 1,25(OH)2D3, 25(OH)D, and 24,25 (OH)2 D in human subjects at 72 hrs after drug administration. Despite previous study did not measure early time points such as author did in this paper, the authors should look at both short-term and long-term changes of the parameters measured in this manuscript. Authors revealed that a large bolus dose could induce high FGF23 which can negatively affect bone. It would be interesting to look at long-term effects because prolonged high FGF23 can cause bone resorption instead of bone accrual. The subject numbers are relative few. The study was performed almost 10 years ago. Bone turnover markers were not measured. The study provided limited guidance for clinical application using cholecalciferol to treat bone osteopenia or osteoporosis. Some minor comments:
1. There is format issue with the Font size are quite different.
2. Line 195, what is “last group” ? Please be more specific.
3. Line 207, “the last dose” means 600, 000 IU?
4. Previous study (PMID: 22739976) does not support FGF23 mediated vitamin D catabolism.
5. ANOVA was not used consistently in the text and figures.
Comments on the Quality of English LanguageEnglish is fine. No major issues were identified.
Author Response
REVIEWER 2
This clinical study investigates cholecalciferol in more physiology dose and bolus dose on the serum 1,25(OH)2D3, 25(OH)D, and 24,25 (OH)2 D in human subjects at 72 hrs after drug administration. Despite previous study did not measure early time points such as author did in this paper, the authors should look at both short-term and long-term changes of the parameters measured in this manuscript. Authors revealed that a large bolus dose could induce high FGF23 which can negatively affect bone. It would be interesting to look at long-term effects because prolonged high FGF23 can cause bone resorption instead of bone accrual. The subject numbers are relative few. The study was performed almost 10 years ago. Bone turnover markers were not measured. The study provided limited guidance for clinical application using cholecalciferol to treat bone osteopenia or osteoporosis. Some minor comments:
We thank the reviewer for the constructive, insightful criticism and suggestions that helped us to clarify some points and improve the quality of our manuscript.
Your suggestion regarding the long-term evaluation of some biochemical parameters is appropriate. However, we would like to emphasize that we have already published some long-term studies concerning vitamin D and its metabolites (see references 19,20,21 of the original manuscript).
Concerning FGF23, we fully agree with your observation. Indeed, we are carrying out a follow-up study investigating the long-term effect of various doses of cholecalciferol on both intact and C-terminal FGF23. Unfortunately, these results at this point in time are not ready for inclusion in this paper.
You are perfectly right that our study was carried out some time ago. The inability to publish our investigation earlier, was due to a number of reasons, not last the COVID pandemic, which has seen our hospital engaged in the frontline for more than 3 years. We sincerely do not believe that this can represent an issue for accepting the paper.
Some of the limitations you addressed have now been included in the manuscript.
Finally, our paper was intended to investigate pathophysiological changes of Vitamin D metabolites following two different doses of cholecalciferol. As such it has no direct clinical application for both treatment or prevention of osteopenia and osteoporosis. However, as highlighted in the text, the increase of 24,25(OH)2D may have some implications.
- There is format issue with the Font size are quite different.
Thanks for picking up this point, that has now been corrected.
- Line 195, what is “last group”? Please be more specific.
This has now been specified, thus avoiding any misunderstanding.
- Line 207, “the last dose” means 600, 000 IU?
Yes. We changed in “This last dose…”
- Previous study (PMID: 22739976) does not support FGF23 mediated vitamin D catabolism.
Once again thanks for this insightful observation. However, we must acknowledge that this field needs more investigations. Some animal studies go in the opposite directions (see for example: Shimada T, Hasegawa H, Yamazaki Y, et al. FGF-23 is a potent regulator of vitamin D metabolism and phosphate homeostasis. J Bone Miner Res. 2004;19(3):429-435).
- ANOVA was not used consistently in the text and figures.
Thanks for your observation. We now better specified in the statistical section which tests we used.
Reviewer 3 Report
Comments and Suggestions for Authors
I appreciate the opportunity to review “Short term effects of escalating doses of cholecalciferol on FGF23 and 24,25(OH)2 vitamin D levels”. This article aims to assess the short-term effect (i.e. 72 hours) of small and large oral dose of cholecalciferol on FGF23 and 24,25(OH)2D in vitamin D deficient individuals, together with the evaluation of three vitamin D metabolic ratios. This dose-response study holds potential for understanding the effects of vitamin D3 bolus doses; however, several areas require revision to enhance the manuscript's clarity, and overall impact. Below are detailed comments and recommendations:
1. The introduction lacks sufficient detail regarding the potential impact of small and large oral doses of cholecalciferol on phosphate homeostasis, which is crucial given the study’s focus on FGF23.
2. Materials and Methods: The section requires a clearer structure with appropriate subheadings for better readability and comprehensibility. Organize the section into distinct subsections: Study Design; Ethical Considerations, and include the approval number; Paricipants (Recruitment, Inclusion Criteria, Exclusion Criteria, Sample Size); Materials and Measurements; Biochemistry.
3. In the Participants section, it is recommended to add a new figure—a flow chart of the selection of study participants.
4. The manuscript does not provide information on sample size calculations, and the recruitment of only 9 patients per group raises concerns about the statistical power. Include detailed calculations of sample size and a power analysis to justify the adequacy of the sample size. Discuss the limitations of statistical power in the context of the small sample size, positioning the study as a preliminary investigation.
5. Statistical Analyses: The statistical section is insufficiently detailed. Specify the statistical tests used, including checks for normality. Provide a rationale for the choice of statistical methods, including any adjustments for multiple comparisons if applicable.
6. Biochemical Measurements: The omission of serum phosphate levels as a biochemical measurement is a gap, considering the study's focus on phosphate-related biomarkers. Authors should justify the exclusion of serum phosphate levels. If serum phosphate data are available, include them in the results to provide a more comprehensive assessment.
7. Given the small total sample size (N=27), the study should be framed as a preliminary investigation throughout the manuscript. Amend the relevant sections to explicitly state that this study is preliminary.
8. Results (line 148): Table 1 lacks statistical analysis for comparing the three groups receiving different doses of cholecalciferol or placebo. Include comparative statistical analyses, detailing any significant differences between groups. Explicitly state the statistical tests used and report p-values.
9. Results (lines 142-147): Although the authors mention a lack of sex differences data stratified by sex are not presented. Include sex-specific data either in the Supplementary Data, and analyze these differences with appropriate statistical tests. This addition will enhance the manuscript's comprehensiveness.
10. Results (lines 167-168): Figure 1 is missing a title, and there is a lack of detail regarding between-group differences and the statistical tests used.
11. Results: Ensure Tables 1 and 2 include detailed statistical analyses, clearly identifying the test used for each comparison.
12. Discussion: The study does not adequately address participant selection in the context of other potential confounding factors such as total dietary vitamin D intake and sunlight exposure. Expand the discussion to consider these factors.
13. At the Discussion section, further emphasize the following themes: implications of this study.
14. At the end of the Discussion section, add a separate section titled "Strengths and Limitations."
15. Conclusions: Conclude with a concise summary of the main findings, emphasizing the potential clinical implications and suggesting avenues for future research.
Comments on the Quality of English Language
The manuscript requires a review for English language quality.
Author Response
REVIEWER 3
I appreciate the opportunity to review “Short term effects of escalating doses of cholecalciferol on FGF23 and 24,25(OH)2 vitamin D levels”. This article aims to assess the short-term effect (i.e. 72 hours) of small and large oral dose of cholecalciferol on FGF23 and 24,25(OH)2D in vitamin D deficient individuals, together with the evaluation of three vitamin D metabolic ratios. This dose-response study holds potential for understanding the effects of vitamin D3 bolus doses; however, several areas require revision to enhance the manuscript's clarity, and overall impact. Below are detailed comments and recommendations:
We thank this reviewer for his positive and constructive comments that helped us to improve the quality of the manuscript. Below you will find our answers that have been also incorporated in the revised version of the manuscript.
- The introduction lacks sufficient detail regarding the potential impact of small and large oral doses of cholecalciferol on phosphate homeostasis, which is crucial given the study’s focus on FGF23.
According to your suggestion, we added an entire paragraph in the introduction with the aim of describing the interrelationships between 1,25(OH)2D and phosphate homeostasis.
- Materials and Methods: The section requires a clearer structure with appropriate subheadings for better readability and comprehensibility. Organize the section into distinct subsections: Study Design; Ethical Considerations, and include the approval number; Paricipants (Recruitment, Inclusion Criteria, Exclusion Criteria, Sample Size); Materials and Measurements; Biochemistry.
According to your suggestion, we better organize the materials and Methods section for better readability and comprehension.
- In the Participants section, it is recommended to add a new figure—a flow chart of the selection of study participants.
Even though, we appreciated your effort to render the manuscript easy to read, we believe that such an information is already present in the main text of the manuscript and, such as, the Figure would appear redundant.
- The manuscript does not provide information on sample size calculations, and the recruitment of only 9 patients per group raises concerns about the statistical power. Include detailed calculations of sample size and a power analysis to justify the adequacy of the sample size. Discuss the limitations of statistical power in the context of the small sample size, positioning the study as a preliminary investigation.
We thank the reviewer for his insightful comments. We would like to emphasize that at the time when the study was carried out there were not papers in the literature reporting short term changes of both 24,25(OH)2D and FGF23 following vitamin D administration. We are thankful for your suggestion about the preliminary nature of the investigation. Accordingly, the title of the manuscript has been changed. In addition, this drawback has been added among the limits of the study.
- Statistical Analyses: The statistical section is insufficiently detailed. Specify the statistical tests used, including checks for normality. Provide a rationale for the choice of statistical methods, including any adjustments for multiple comparisons if applicable.
Once again, we thank you for giving us the possibility of improving our manuscript by including the information missing.
- Biochemical Measurements: The omission of serum phosphate levels as a biochemical measurement is a gap, considering the study's focus on phosphate-related biomarkers. Authors should justify the exclusion of serum phosphate levels. If serum phosphate data are available, include them in the results to provide a more comprehensive assessment.
We acknowledge that omission of serum phosphate determination constitutes a drawback. This has been included among the limitations of the paper. However, as you can also see from our previous answer to referee # 1, we are carrying out another similar study evaluating both short- and long-term changes following different doses of cholecalciferol. The determination of serum phosphate is included in this new investigation, also to fill this gap. However, we cannot report these data in the present investigation.
- Given the small total sample size (N=27), the study should be framed as a preliminary investigation throughout the manuscript. Amend the relevant sections to explicitly state that this study is preliminary.
We really thank this reviewer for this smart suggestion that has now been included both in the title and, when needed, in different parts of the manuscript.
- Results (line 148): Table 1 lacks statistical analysis for comparing the three groups receiving different doses of cholecalciferol or placebo. Include comparative statistical analyses, detailing any significant differences between groups. Explicitly state the statistical tests used and report p-values.
According to your suggestion, a more extensive statistical description has been added in the pertinent section. Furthermore, ANOVA p values have been added in Table 1. We would like to emphasize that there were not statistically significant differences among the three groups studied regarding baseline parameters.
- Results (lines 142-147): Although the authors mention a lack of sex differences data stratified by sex are not presented. Include sex-specific data either in the Supplementary Data, and analyze these differences with appropriate statistical tests. This addition will enhance the manuscript's comprehensiveness.
Below you will find a table with the number of males and females investigated, within each group. Given the small numbers of both males and females, it is not possible to analyze the biochemical differences between the groups. In the discussion we added this aspect as a limitation of the paper. We prefer not to include this table as a supplementary one, given the limited information presented.
|
Dose |
Females |
Males |
|
25,000 I.U. |
6 |
2 |
|
600,000 I.U. |
7 |
2 |
|
Placebo |
3 |
6 |
- Results (lines 167-168): Figure 1 is missing a title, and there is a lack of detail regarding between-group differences and the statistical tests used.
According to your suggestion, a title has been added to Figure 1. The statistical analyses have been updated, even though the information was already present in the text.
- Results: Ensure Tables 1 and 2 include detailed statistical analyses, clearly identifying the test used for each comparison.
The statistical tests used for the description of Table 1 and 2 have been better described in the statistical section.
- Discussion: The study does not adequately address participant selection in the context of other potential confounding factors such as total dietary vitamin D intake and sunlight exposure. Expand the discussion to consider these factors.
The issue raised has been addressed in the Material and Methods section. Please consider that the length duration was only three days and that patients were recruited in a very short period of time.
- At the Discussion section, further emphasize the following themes: implications of this study.
This aspect has also been added in the discussion.
- At the end of the Discussion section, add a separate section titled "Strengths and Limitations."
A separate section about strengths and limitations has been added.
- Conclusions: Conclude with a concise summary of the main findings, emphasizing the potential clinical implications and suggesting avenues for future research.
We changed the conclusions with a new heading emphasizing clinical implications and suggesting future research activities.
Reviewer 4 Report
Comments and Suggestions for Authors
The manuscript presents the effects of escalating doses of cholecalciferol on FGF23 and 24,25(OH)2 vitamin D levels. Is an interesting study which can present interest for this journal s readers. I recommend the publication after a major revision.
As comments/suggestions:
1. I believe that the number of patients included in the study (27) divided into 3 groups is too small.
2.How relevant is this study considering that there are both women and men participating in the study?
3.Why was the dosage of the inactive hydroxylated metabolite 24,25(OH)2 chosen, considering that following the metabolism of vitamin D there are other inactive hydroxylated metabolites?
Author Response
REVIEWER 4
The manuscript presents the effects of escalating doses of cholecalciferol on FGF23 and 24,25(OH)2 vitamin D levels. Is an interesting study which can present interest for this journal s readers. I recommend the publication after a major revision.
We thank the reviewer for his comment.
As comments/suggestions:
- I believe that the number of patients included in the study (27) divided into 3 groups is too small.
We completely agree with you. Indeed, this was a pilot study. To better define this point, also highlighted by another reviewer, we modified the title accordingly.
2.How relevant is this study considering that there are both women and men participating in the study?
Thanks for this comment. We believe that the inclusion of both men and women in the investigation represents a strength of the manuscript so that the results obtained can be generalized and applied to both sexes.
3.Why was the dosage of the inactive hydroxylated metabolite 24,25(OH)2 chosen, considering that following the metabolism of vitamin D there are other inactive hydroxylated metabolites?
When the study was carried out, there was a lot of interest in this metabolite that has now been renewed because of possible links with muscles and falls, as has been emphasized in the discussion. A new investigation we are carrying out also address your observation.
Round 2
Reviewer 2 Report
Comments and Suggestions for Authors
Authors have revised the manuscript, but did not add any new experiments or data. I still have the following comments for the revised manuscript.
1. Line 155-156, “ To 500 L serum, we added 50L”, are these volume correct?
2. Line 186-188, “Anova” should be all capital letter.
3. Line 209, there is “last group” not clarified, just be consistent using 600,000u group.
4. Figure 1 legend should be all below the figure, there is no “A,B,C” in the figure, but legend has A,B,C. Please be consistent.
5. Line 268, authors said “bolus dose”, are they indicate 600,000IU dose? I guess mention the dose directly is more straight forward, here in this study both doses only used 1 time, all bolus dose.
Author Response
Authors have revised the manuscript, but did not add any new experiments or data. I still have the following comments for the revised manuscript.
- Line 155-156, “ To 500 L serum, we added 50L”, are these volume correct?
Thanks for your observation. The paragraph concerning assay of 24,24 (OH)2D has been completely substituted, so that any possible mistake has been eliminated.
- Line 186-188, “Anova” should be all capital letter.
This has been corrected everywhere.
- Line 209, there is “last group” not clarified, just be consistent using 600,000u group.
This has been better specified.
- Figure 1 legend should be all below the figure, there is no “A,B,C” in the figure, but legend has A,B,C. Please be consistent.
Thank you, once again for picking up this issue, that has been corrected.
- Line 268, authors said “bolus dose”, are they indicate 600,000IU dose? I guess mention the dose directly is more straight forward, here in this study both doses only used 1 time, all bolus dose.
This has been specified.
We thank this reviewer for another round of constructive suggestions that improved our manuscript. We hope that you can now consider our paper suitable for definitive publication
Reviewer 3 Report
Comments and Suggestions for Authors
The authors have answered all queries appropriately. The manuscript has been improved. I have no further comments and suggestions.
Author Response
The authors have answered all queries appropriately. The manuscript has been improved. I have no further comments and suggestions.
We thank this reviewer for his constructive suggestions that improved our manuscript.
Reviewer 4 Report
Comments and Suggestions for Authors
The authors modified the manuscript following the suggestions of the reviewer.
In my opinion the article can be published in present form
Author Response
The authors modified the manuscript following the suggestions of the reviewer.
In my opinion the article can be published in present form.
We thank this reviewer for his constructive suggestions that improved our manuscript.